# Harm Perceptions and Beliefs about Potential Modified Risk Tobacco Products

**DOI:** 10.3390/ijerph18020576

**Published:** 2021-01-12

**Authors:** Jennifer C. Morgan, Joseph N. Cappella

**Affiliations:** Annenberg School for Communication, University of Pennsylvania, Philadelphia, PA 19104, USA; JCappella@asc.upenn.edu

**Keywords:** alternate tobacco products, e-cigarettes, modified risk tobacco products

## Abstract

Under US law, tobacco products may be authorized to claim lower exposure to chemicals, or lower risk of health harms. We sought to examine the harm perceptions and beliefs about potential modified risk tobacco products (MRTPs). We recruited 864 adult current and former smokers in August 2019. Participants read a paragraph describing the potential for the FDA to authorize MRTPs and a brief description of MRTPs. The most endorsed beliefs for each product were that they contained nicotine and that they were risky. Believing that e-cigarettes can help smokers quit smoking, that they tasted good, and looked cool were associated with greater odds of intending to try e-cigarettes after controlling for demographic and use factors. For snus, the beliefs that the product was not addictive and tasted good were associated with increased odds of intending to try snus. The beliefs that heated tobacco would taste good and would be a good quit aid was associated with increased odds of intentions to try heated tobacco products. Understanding what the public believes about products currently or potentially authorized to be marketed as modified risk tobacco products can inform communication efforts.

## 1. Introduction

There is a growing recognition among tobacco control experts that tobacco products pose different levels of risk which exist along a “continuum of risk” [1,2]. Snus (a low nitrosamine type of moist snuff), and e-cigarettes pose significantly fewer health risks to individual users compared to cigarettes, though they are not without risk [2,3,4,5]. While switching to products lower on the risk continuum may have the potential for harm reduction among those smokers who are unable or unwilling to quit cigarettes; research indicates that users’ perceptions about the comparative risks of tobacco products are often inconsistent with the continuum of risk. Many believe smokeless tobacco (SLT) and e-cigarettes are as harmful or more harmful than tobacco cigarettes [6,7,8]. Many tobacco control experts have called for more accurate communication about the risks of such products relative to cigarettes [8,9,10]. Smokers may be receptive to such communications, with about half reporting they would be interested in using a tobacco product that claimed to be less harmful than other tobacco products [11,12].

In response to the health concerns about the harmful health effects of smoking, the tobacco industry began marketing new tobacco products as less harmful compared to traditional cigarettes [13,14]. This history can be traced from the recent efforts to advertise e-cigarettes as a harm reduction alternative to cigarettes, before becoming regulated by the FDA, [15,16] back to falsely claiming that filtered, “low tar,” and “light” cigarettes were less harmful than regular cigarettes [17,18,19]. After decades of misleading reduced risk claims, [20] the 2009 US Family Smoking Prevention and Tobacco Control Act (Tobacco Control Act) provided a regulatory framework in which tobacco companies could introduce and market tobacco products with lower exposure or risk claims only after a review and obtaining a marketing order from the US Food and Drug Administration (FDA) [21]. Under the law, products with these claims are “modified risk tobacco products” (MRTPs), defined as products “sold or distributed for use to reduce harm or the risk of tobacco-related disease associated with commercially marketed tobacco products” [21]. To date, applications from five brands, including a reduced-nicotine cigarette (22nd century), an electronic heated tobacco product (i.e., IQOS), and three SLT products (i.e., General Snus, Camel Snus, and Copenhagen) have been submitted. The FDA issued the first modified risk order to Swedish Match authorizing a claim that “Using General Snus instead of cigarettes puts you at lower risk of mouth cancer, heart disease, lung cancer, stroke, emphysema, and chronic bronchitis” in October 2019 [22] In July 2020, FDA authorized an exposure modification claim for IQOS, which states that switching completely from cigarettes to IQOS can “significantly reduce your body’s exposure to harmful or potentially harmful chemicals” [23].

The potential impact of MRTP claims on population health may in part depend on whether smokers are exposed to these claims, how they view MRTP claims on their own and in comparison to other tobacco products, and if they perceive them to be salient and truthful [24,25,26]. Attitudes and beliefs about tobacco products can be considered as either comparative or absolute [27]. Absolute measures of beliefs and perceptions focus on each tobacco product without a comparison product, where comparative measures include a reference to a specific tobacco product, typically cigarettes [28]. Using a theory of reasoned action lens to approach this topic, how the population reacts to these claims will be in part determined by their beliefs, including misbeliefs, about using these products [29]. Health mass media campaigns that follow principles of effective campaigns can have moderate effects on health knowledge, beliefs, attitudes, and behaviors [30,31]. One of these principles is helping campaign designers understand the nature of the behavior, for a campaign can only be effective if the beliefs targeted by the campaign impact the intended behavior.

The purpose of this this study is to provide valuable baseline data about smokers’ and former smokers’ beliefs about three products that currently have been or have the potential to be authorized as modified risk tobacco products. Specifically, we sought to understand harm perceptions and beliefs about products with the potential to be authorized as MRPTs; examine differences in beliefs about using MRTPs and beliefs about MRTPs in comparison to cigarettes; how MRTP user status impacts the rating of beliefs; and which beliefs are associated with intentions to use MRTPs.

## 2. Methods

We recruited 864 adult current and former smokers through Dynata to complete an online survey in August 2019, before any products had been authorized as MRTPs and before e-cigarette or vaping product use-associated lung injury (EVALI) was in the news. Participants were considered current smokers if they had smoked at least 100 cigarettes in their lifetime and currently smoke every day or some days, and former smokers if they had smoked at least 100 cigarettes in their lifetime, and currently did not smoke at all [32]. Additionally, participants could not have participated in more than two online surveys about cigarette smoking or other tobacco products in the last three months.

Participants included 450 men and 414 women, with a mean age of 47.5 years. A little more than half of the participants were current smokers or ever-users (i.e., current or former users) of e-cigarettes. Participants were diverse in race, education, and income, though slightly more educated than the population of US smokers (Table 1) [32].

### 2.1. Procedures

Participants completed an eligibility survey, answered demographic survey questions, and information about their current smoking including the Fagerstrom scale [33], and the contemplation ladder [34]. Then, participants read a paragraph describing the potential for the FDA to authorize modified risk tobacco products (MRTPs) and a brief description of MRTPs. Participants were randomly assigned to either the control condition, in which they read a description about heat-not burn tobacco, snus, and e-cigarettes, presented in random order (Table 2); or to the corporate social responsibility condition, in which they read the generic description, plus a corporate responsibility statement crafted using press releases and text from IQOS, General Snus, or JUUL’s website respectively. There were no differences between the conditions, so the conditions were collapsed for further analysis.

After reading descriptions of MRTPs, and a description of e-cigarettes, snus, and heated tobacco, participants answered questions about awareness, harm perceptions, use, likelihood of use, beliefs, and intentions to use the three products that have potential for, or have already received, authorization for modified risk tobacco product claims. The belief questions were worded using three different variations and participants were randomly assigned one wording variation per product using a Latin square design. The [blinded for review purposes] institutional review board approved the procedures.

### 2.2. Measures

#### 2.2.1. Awareness

Participants were provided with brief descriptions of e-cigarettes, snus, and heated tobacco and then indicated if they had ever heard of each product. Those who answered yes were considered aware of the product [35,36].

#### 2.2.2. Harm Perceptions

Participants indicated how harmful they thought e-cigarettes, snus, and heated tobacco were to their health on a scale of not at all harmful (coded as 1), to extremely harmful (coded as 4) [35].

#### 2.2.3. Ever-Use and Frequency of Use

Participants who were aware of a product were asked if they had ever used the product even one or two times. Those who answered that they were unaware, or who had never used the product were coded as never-users. Ever-users were then asked “In the past 30 days, on how many days did you use [product]?” Ever-users who answered that they had not used the product in the last 30 days were coded as former users, those who had used the product in the last 30 days were coded as current users. The heated tobacco and snus items were adapted from the PATH survey item about e-cigarettes [35,36].

#### 2.2.4. Likelihood of Use

All participants were asked “If one of your best friends were to offer you a [product], would you use it?” The 4-point response scale ranged from definitely no (coded as 1) to definitely yes (coded as 4) [35,36].

#### 2.2.5. MRTP Beliefs

The survey first assessed beliefs with an open-ended question asking “What comes to mind when you think about [product]?” Participants were instructed to list up to 5 thoughts about the products. After completing the awareness, harm perceptions, use, and open-ended beliefs for each of the three products, the survey assessed 15 different beliefs about e-cigarettes, snus and heated: tasting good, feeling harsh, being odorless, being easy to use, looking cool, making second hand smoke, being risky, not being addictive, having long term health benefits, causing lung damage, containing nicotine, helping smokers quit, untrustworthy science about the product, appealing to kids, and being expensive (Appendix A). The decision to focus on these beliefs was based on prior qualitative research, surveys on salient beliefs, and examining the MRTP applications and materials made publicly available [12,25,37,38,39,40,41,42,43,44,45]. The 5-point response scale ranged from strongly disagree (coded as 1) to strongly agree (coded as 5). The survey assessed these beliefs using three wording variations: a comparison to cigarettes, a self-referent, and an absolute statement. Using a Latin square design, participants were asked about their beliefs for each product using a different wording condition. Group A answered questions about heated tobacco with the comparison wording, the absolute was used for snus, and the self-referent was used for e-cigarettes. Group B answered e-cigarette beliefs with the absolute wording, self-referent wording was used for heated, and comparison wording was used to assess snus beliefs. For group C, beliefs about snus were assessed using the self-referent wording, e-cigarette beliefs were assessed using the comparison wording, and heated tobacco beliefs items used the absolute wording. This design insures that—despite three sets of belief wordings—in each group the respondent always responds to a different wording-product combination without repeating the wording or the product.

Spearman ranked correlations between the absolute wording and the self-referent wording were very high (ρ (rho) = 0.97 to 0.98, *p* < 0.001) for each of the three products indicating that participants did not meaningfully differentiate between the two wordings. The correlation remained high even after subgroup analysis comparing demographic characteristics, smoking status, and user characteristics. The self-referent and absolute wording conditions were collapsed in subsequent analyses.

#### 2.2.6. Intentions to Use MRTP Products

The survey asked if the participant thought they would use e-cigarettes, snus, or heated tobacco in the next year. The 4-point response scale ranged from definitely no (coded as 1) to definitely yes (coded as 4) [35,36].

### 2.3. Statistical Analysis

Stata version 15 [46] was used to conduct Spearman rank correlations, and descriptive analysis to obtain means and standard deviations. Intention to use potential MRTPs was dichotomized in order to compare responses of “definitely no” (coded as 0), to other responses (coded as 1). We specified multiple logistic regression models in Stata version 15 to assess the intention to use each of the alternative tobacco products. Each model includes scores for each of the 15 beliefs. Each model also controlled for the individual-level covariates including ever use of at least 1 of the products, current smoking status, and sociodemographic characteristics: education, ethnicity, race, and age. We conducted a likelihood-ratio test to evaluate the difference between the model with only the individual-level covariates, and the full model with all 15 beliefs and all the individual-level covariates.

## 3. Results

### 3.1. Awareness and Use of Potentially Modified Risk Tobacco Products

E-cigarettes were the most well-known product of the three potentially modified risk tobacco products we asked about, with 97% of participants reporting they had heard of them. Only 12.5% of the participants had heard of heated tobacco (Table 3). Fifty four percent of participants had ever used at least one of the products, the overwhelming majority of current users and former MRTP users were e-cigarette users.

### 3.2. Harm Perceptions and Likelihood of Use for Potentially Modified Risk Tobacco Products

Mean perceptions of harm were 2.60 (sd = 0.92) for heated tobacco, 2.67 (sd = 0.90) for e-cigarettes, and 2.81 (sd = 0.90) for snus (F = 18.93, *p* < 0.01), post-hoc analysis indicated that the participants had higher harm perceptions for snus compared to each of the other two products (*p* > 0.05). The likelihood to use was lowest for snus with a mean of 1.5 (sd = 0.84), for heated tobacco the mean was 1.80 (sd = 0.95), and for e-cigarettes the mean likelihood to use was highest at 2.1 (sd = 1.13) (F = 76.7, *p* < 0.01). Post-hoc analysis indicated that the mean likelihood to use each product was significantly different from the other two products (*p* < 0.01).

### 3.3. E-Cigarette Beliefs

#### 3.3.1. Beliefs about Using E-Cigarettes

The three most endorsed beliefs about using e-cigarettes were that they contained nicotine (M = 3.98, SD = 0.93), they were risky (M = 3.89, SD = 1.01), and that they caused lung damage (M = 3.87, SD = 0.95). Participants were least likely to endorse the beliefs that e-cigarettes had long-term health benefits (M = 2.56, SD = 1.40), that they were cool (M = 2.38, SD = 1.22), and that there were not addictive (M = 2.23, SD = 1.18; Table 4).

#### 3.3.2. Beliefs about E-Cigarettes in Comparison to Cigarettes

When a comparison to cigarettes was invoked, participants most commonly endorsed the belief that e-cigarettes appealed to kids (M = 3.57, SD = 1.18), that the science about e-cigarettes could not be trusted (M = 3.44, SD = 1.14), and that e-cigarettes were expensive (M = 3.29, SD = 1.03). Similar to their beliefs about using e-cigarettes, participants were least likely to endorse the beliefs that e-cigarettes were cool (M = 2.59, SD = 1.17), had long-term health benefits (M = 2.51, SD = 1.33), and were not addictive when compared to cigarettes (M = 2.29, SD = 1.15; Table 4).

#### 3.3.3. Differences between MRTP User Groups

The correlation of the rankings of beliefs about using e-cigarettes between never, former, and current was high (Rho > 0.73; *p* < 0.001) and not independent from each other. The correlation of the rankings of beliefs about e-cigarettes compared to cigarettes between never and current users (Rho= −0.09; *p* = 0.74) and former users and current users (Rho= 0.41; *p* = 0.13) was low, and independent of each other. Current users most highly rated belief was that the e-cigarettes taste good compared to cigarettes (mean = 3.8, SD= 1.03), among former-users it came in the middle, rated 5th highest (M= 3.10, SD = 1.20), and among never users it was rated 10th (m = 2.86, SD = 0.96). Current users lowest rated belief was that e-cigarettes were risky compared to cigarettes (M= 2.58, SD = 1.08), former users rated it 6th (M = 3.06, SD = 1.10) and never users rated it 5th (M = 3.27, SD = 3.27). Former users and never-users’ lowest rated belief was that e-cigarettes were not addictive compared to cigarettes (M_former_ = 1.92, SD = 0.96; M_never_ = 2.03, SD = 0.99), whereas current users rated it 10th (M = 2.93, SD = 1.23). Never users highest rated belief is that the science about e-cigarettes is untrustworthy compared to cigarettes (M = 3.82, SD = 1.01) compared to current users, who rated it 9th highest (M = 3.04, SD = 1.17), and former users who rated it 4th highest (M = 3.23, SD = 1.11).

### 3.4. Snus Beliefs

#### 3.4.1. Beliefs about Using Snus

The three most endorsed beliefs about using snus were that it contains nicotine (M = 3.98, SD = 0.93), that it is risky (M = 3.95, SD = 0.97), and that it does not make second-hand smoke (M = 3.60, SD = 1.09). Participants were least likely to endorse the beliefs that snus tasted good (M = 2.48, SD = 1.08), that it was not addictive (M = 2.20, SD = 1.16), and that it was cool (M = 2.05, SD = 1.14; Table 4).

#### 3.4.2. Beliefs about Snus in Comparison to Cigarettes

When a comparison to cigarettes was invoked, participants most commonly endorsed the belief that snus produced no second-hand smoke (M = 3.73, SD = 1.05), that the science about snus was untrustworthy (M = 3.25, SD =1.08), and that snus was expensive compared to cigarettes (M = 3.23, SD = 1.08). Like their beliefs about using snus, participants were least likely to endorse the beliefs that snus cool (M = 2.28, SD = 1.02), was not addictive compared to cigarettes (M = 2.40, SD = 1.11) and would help smokers quit smoking (M = 2.53, SD = 1.07; Table 4).

#### 3.4.3. Differences between MRTP User Groups

The rankings of beliefs between users of MRTPs was high for snus (Rhos_self_ > 0.89; *p* < 0.001; Rhos_comparison_ > 0.74; *p* < 0.01), and not independent of one another.

### 3.5. Heated Tobacco Beliefs

#### 3.5.1. Beliefs about Using Heated Tobacco

The three most endorsed beliefs about using heated tobacco was that it contains nicotine (M = 3.80, SD = 0.96), that it is risky (M = 3.76, SD = 0.99), and that it causes lung damage (M = 3.63, SD = 0.96). Participants were least likely to endorse the beliefs that heated tobacco would help smokers quit (M = 2.48, SD = 1.08), that it was not addictive (M = 2.20, SD = 1.16), and that it was cool (M = 2.30, SD = 1.18; Table 4).

#### 3.5.2. Beliefs about Heated Tobacco in Comparison to Cigarettes

When a comparison to cigarettes was invoked, participants most commonly endorsed the belief that heated tobacco was expensive (M = 3.43, SD = 0.83), that the science about heated tobacco was untrustworthy (M = 3.41, SD = 0.98), and that heated tobacco products were appealing to kids compared to cigarettes (M = 3.30, SD =1.05). Like their beliefs about using heated tobacco products, participants were least likely to endorse the beliefs that heated tobacco had long-term benefits (M = 2.66, SD = 1.15), were not addictive compared to cigarettes (M = 2.60, SD = 1.12) and would help smokers quit smoking (M = 2.66, SD = 1.02; Table 4).

#### 3.5.3. Differences between MRTP User Groups

The rankings of beliefs between never, former, and current users of MRTPs were high for heated tobacco products (Rhos_self_ > 0.96; *p* < 0.001), and not independent of each other. The correlation of the rankings of beliefs about heated tobacco compared to cigarettes between current and never users (Rho = 0.45, *p* = 0.09) and current and former users (Rho = 0.69, *p* = 0.004) were moderate. For the belief that heated tobacco would not create second-hand smoke compared to cigarettes, current users and former users rated it 2nd and 3rd highest, respectively (M_current_ = 3.33, SD = 1.01; M_former_ = 3.21, SD = 1.00). In comparison, never-users rated the belief 9th highest (M = 2.84, SD = 0.89). Current users rated the belief that heated tobacco would cause lung damage 13th (M = 3.04, SD = 1.04) compared to former MRTP users who rated it 7th (M = 3.00, SD = 0.89) and never MRTP users who rated it 4th (M = 3.26, SD = 0.82). Both former and never MRTP users rated the belief that heated tobacco was risky compared to cigarettes 5th (M_never_ = 3.19, SD = 0.85; M_former_ = 3.08, SD = 0.82), whereas current MRTP users rated it 10th (M = 3.06, SD = 1.04).

### 3.6. Beliefs from Open Ended Responses

Participants were instructed to list up to 5 thoughts about the products. This was done to ensure that we did not miss any widespread beliefs in our set of 15 close-ended responses that followed. After discarding nonsensical responses (e.g., hdakhjk), and don’t know responses (e.g., “don’t know,” “none,” “n/a”), we coded responses as one of the 15 close ended beliefs, or as a new response. New responses were then coded into new belief groupings. When participants were asked to list their beliefs about snus and heated tobacco, the most common responses were about non-lung health concerns or benefits, the most common open-ended responses about e-cigarettes were about risk and safety. Among the beliefs not asked about in the close-ended portion, participants listed general affective reactions (e.g., gross, eww) for all three products. For heated tobacco and snus, participants commonly listed associations they made with the product (e.g., baseball, chew, hookah, vaping) (Table 5).

### 3.7. Intentions to Use Potentially Modified Risk Tobacco Products

For all products and wording conditions, including beliefs increased the models’ fit compared to models that only included the demographic control variables.

#### 3.7.1. E-Cigarettes

Believing that e-cigarettes can help smokers quit smoking (OR = 1.61), that they tasted good (OR = 1.56), and looked cool (OR = 1.42) were associated with greater odds of intending to try e-cigarettes after controlling for demographic and use factors. Believing that the science about e-cigarettes was untrustworthy was associated with lower odds of intending to try e-cigarettes (OR = 0.61).

#### 3.7.2. Snus

The belief that the product was not addictive (OR = 2.07) and tasted good (OR = 1.45) were associated with increased odds of intending to try snus. Believing that the science about snus was untrustworthy (OR = 0.59) and that the product was risky (OR = 0.63) was associated with lower odds of intending to try snus.

#### 3.7.3. Heated Tobacco

The belief that heated tobacco would taste good (OR = 2.92) and would be a good quit aid (OR = 1.72) was associated with increased odds of intentions to try heat-not burn tobacco products. Believing that heated was expensive was associated with lower odds of intending to try (OR = 0.67; Table 6).

## 4. Discussion

In order for modified risk tobacco products to live up to their harm reduction potential, two things must be true simultaneously: MTRPs cannot entice nonsmokers to start using them when they otherwise would have never used a tobacco product, and they cannot spur smokers to switch to them when they otherwise would have quit tobacco completely. They will have the greatest public health impact if they move people down the risk continuum who would have otherwise remained a user of the riskier product. This maybe particularly salient as the US moves to put pictorial tobacco warnings on cigarette packs. As these warnings increase intentions to quit smoking, this may increase interest in switching to a less harmful product [47].

The results of this study demonstrate that adult smokers and former smokers have different awareness, harm perceptions, and beliefs the three products they assessed. This is consistent with prior cross-sectional research showing that the public views some products as more harmful than others [6,48]. This paper adds to this literature by including perceptions and beliefs about heated tobacco alongside e-cigarettes and snus. This sample of current and former smokers perceived heated tobacco as less harmful than snus, but not significantly different from e-cigarettes. Prior research with Canadian respondents found that over half of the respondents perceived IQOS as equally or more harmful than e-cigarettes [49]. Given the importance of moving people down the risk continuum, the differences in beliefs about using the products and the products when compared to cigarettes are of particular importance and interest. Across all three products, the most endorsed beliefs were that the products contained nicotine, which is true, and that they are risky, and the least endorsed beliefs all included that the products were not addictive. The participants overall, did not demonstrate any glaring misbeliefs about using any of the three products. When a comparison to cigarettes was invoked, the most endorsed beliefs were about trustworthiness of the science, appeal of the product to kids, and the cost.

The correlations of the rankings between different user groups were high for beliefs about using all three products. However, when the comparison to cigarettes was invoked, participants with different MRTP use history differ in their endorsements of the 15 beliefs about e-cigarettes and heated tobacco. Current users of MRTPs were less likely to endorse beliefs about risk and lung damage compared to never and former users. The fact that there are differences between never, former, and current users is consistent with research demonstrating the role of involvement on message processing and attitude change [50]. Because current users have more of a stake in the safety of these products than never users, they are likely more attentive to the information, and as a result are affected in different ways, they also are motivated to believe that the products are safe, since they are currently using them.

One of the limitations of this study is that it was a convenience sample that cannot be generalized to the larger US population. While the demographics are similar to those of US smokers, our population was slightly more educated than the population of US smokers. Another limitation is that it is a cross-sectional study and only captures these beliefs at a very specific point in time. As previously mentioned, these data were collected before EVALI news was widespread, and before a global pandemic featuring a respiratory illness (COVID-19) interrupted everyone’s daily lives and routines. While it is encouraging that some of our findings are similar to other large nationally representative longitudinal studies, [6,36,39,49] replicability of these findings with a larger and longitudinal sample would be ideal.

## 5. Conclusions

For all three products, believing that MRTPs taste good and can be used as a quit aid are associated with greater intentions to try MRTPs. This is consistent with prior work which found that young adults who held a baseline belief that e-cigarettes could help people quit were more likely to report experimenting with e-cigarettes at follow-up [39]. Beliefs that the science about the product was untrustworthy was associated with reduced intentions to try the product. However, each product had its own unique sets of beliefs associated with trying the products. This paper contributes to the growing literature about beliefs and perceptions of e-cigarettes, snus, and heated tobacco. It contributes to our understanding of what the public believes about products currently or potentially authorized to be marketed as modified risk tobacco products. This understanding can inform the planning and development stages of communication campaigns.

## Figures and Tables

**Table 1 ijerph-18-00576-t001:** Participant characteristics.

Characteristic	Participants (*n* = 864)
	*n* (%)
Mean age (SD)	47.5 (17.6)
Gender	
Male	450 (52)
Female	414 (48)
Hispanic	59 (7)
Race	
White	732 (85)
Black or African American	81 (9)
Asian	14 (2)
American Indian, Alaska Native, or Native Hawaiian	34 (4)
Education	
High school or less	283 (33)
Some college	201 (23)
College or higher	380 (44)
Income	
Less than $25,000	187 (22)
Between $25,000 and $49,999	213 (25)
Between $50,000 and $74,999	164 (19)
Between $75,000 and $99,999	126 (15)
Between $100,000 and $149,999	117 (14)
$150,000 or more	55 (6)
Smoking status	
Current smoker	463 (54)
Former smoker	402 (46)
MRTP user status	
Current user	288 (33)
Former user	180 (21)
Never user	393 (45)

Note: Missing data range from 0 to 0.6%.

**Table 2 ijerph-18-00576-t002:** Tobacco products currently or with the potential to receive modified risk tobacco products (MRTP) authorization and the product descriptions.

Product	Description
E-cigarettes	Electronic or e-cigarettes and other vaping devices, such as pod mods and vape pens, are battery-powered devices that produce vapor instead of smoke. Some look like cigarettes, some are very colorful, and some have tanks that contain liquid. Popular brands of e-cigarettes include JUUL, Blu, Vuse, NJOY, and Flavor Vapes.
Snus	Snus is a type of moist tobacco. It comes in a small pouch that goes under your lip. It does not result in the need for spitting. Popular brands include General snus, Camel Snus, and Swedish Match.
Heated tobacco	Heat-not-burn is a type of tobacco product. It heats the tobacco just enough to release a small amount of smoke without burning the tobacco. Popular brands include IQOS.

**Table 3 ijerph-18-00576-t003:** Awareness and use of potentially modified risk tobacco products (*n* = 864).

	E-Cigarettes	Snus	Heated Tobacco
	*n* (%)		
Aware of product	839 (97)	488 (56)	107 (12)
User status			
Current user	284 (33)	45 (5)	50 (6)
Former user	168 (19)	58 (7)	5 (1)
Never user	412 (48)	758 (88)	806 (93)
Past 30-day use of ever users mean (sd)	6.9 (10.0)	2.0 (3.6)	4.6 (6.3)

Note: Missing data range from 0 to 0.6%.

**Table 4 ijerph-18-00576-t004:** Mean belief endorsement for the 15 close-ended belief items about e-cigarettes, snus, and heated tobacco *n* = 864.

	E-Cigarette Self	E-Cigarette Comparison	Snus Self	Snus Comparison	Heated Tobacco Self	Heated Tobacco Comparison
	Mean (SD)					
Taste good	3.09 (1.10)	3.21 (1.11)	2.48 (1.08)	2.67 (1.05)	2.78 (0.96)	2.85 (0.84)
Feel harsh	3.14 (0.94)	2.88 (0.95)	3.45 (0.96)	3.12 (0.86)	3.35 (0.86)	3.06 (0.79)
Odorless	2.69 (1.09)	2.83 (1.16)	2.96 (1.05)	3.13 (1.00)	2.68 (0.94)	2.81 (0.94)
Easy	3.61 (0.98)	3.16 (1.10)	3.19 (1.03)	3.17 (1.00)	3.01 (0.94)	2.96 (0.93)
Cool	2.38 (1.22)	2.59 (1.17)	2.06 (1.14)	2.28 (1.02)	2.30 (1.18)	2.67 (1.04)
No second-hand smoke	2.80 (1.13)	3.23 (1.22)	3.60 (1.09)	3.73 (1.05)	2.69 (1.01)	3.08 (0.98)
Risky	3.89 (1.01)	3.01 (1.04)	3.95 (0.97)	3.22 (0.93)	3.76 (0.99)	3.12 (0.91)
Not addictive	2.23 (1.18)	2.29 (1.15)	2.20 (1.16)	2.40 (1.11)	2.35 (1.09)	2.60 (1.12)
Health benefits	2.56 (1.40)	2.51 (1.33)	2.52 (1.38)	2.70 (1.32)	2.59 (1.31)	2.66 (1.15)
Lung damage	3.87 (0.95)	3.08 (1.12)	3.19 (1.11)	2.89 (1.00)	3.63 (0.96)	3.13 (0.92)
Nicotine	3.98 (0.93)	2.97 (0.91)	3.98 (0.93)	3.10 (0.80)	3.80 (0.96)	3.07 (0.78)
Quit	2.69 (1.19)	2.86 (1.22)	2.56 (1.05)	2.53 (1.07)	2.48 (1.01)	2.66 (1.02)
Untrustworthy sci	3.50 (1.08)	3.44 (1.14)	3.41 (1.05)	3.25 (1.08)	3.45 (1.09)	3.41 (0.98)
Kid appeal	3.68 (1.08)	3.57 (1.18)	2.89 (1.21)	2.96 (1.07)	3.04 (1.12)	3.30 (1.05)
Expensive	3.70 (1.05)	3.29 (1.03)	3.58 (0.96)	3.23 (0.88)	3.56 (0.95)	3.43 (0.83)

Note. Belief scale was from strongly disagree (coded as 1) to strongly agree (coded as 5). The same participants did not answer both wording variations for the same product.

**Table 5 ijerph-18-00576-t005:** The top five most common open-ended response categories and example statements for three types of potentially modified risk tobacco product.

	Snus		Heated Tobacco		E-Cigarettes	
	Category	Example Statements	Category	Example Statements	Category	Example Statements
1	Non-lung health concerns or benefits (*n* = 452)	“Bad for gums” “mouth cancer” “unhealthy”	Non-lung health concerns or benefits (*n* = 201)	“causes cancer,” “causes diseases,” “harmful” “it still isn’t good for your body” “death”	Risk or safety (*n* = 217)	“Dangerous,” “A little safer,” “Not safe,” “Harmful,” “Might blow up,” “potentially completely harmless”
2	Affective reactions * (*n* = 278)	“bad,” “good,” “great,” ew,” “gross” “yuck”	Affective reactions * (*n* = 110)	“bad,” “disgusting” “enjoyable,” “ewww,” “good”	Non-lung health concerns or benefits (*n* = 183)	“Deadly,” “Health concerns,” “Bad for you,” “can still cause cancer,” “brain damage,” “healthier”
3	Associations * (*n* = 134)	“chew,” “dip,” “guys,” “baseball,” “spitting,” “tobacco”	Risk or safety (*n* = 108)	“it sounds a little safer than a cigarette,” “it could be less risky than smoking,” “harmful,” “harmless”	Affective reactions * (*n* = 115)	“awesome,” “bad,” “fun,” “good,” “not good,” “gross,” “nasty”
4	Risk or safety (*n* = 82)	“safer than cigarettes,” “very harmful,” “just another unhealthy tobacco product”	Association * (*n* = 93)	“clean,” “hookah,” “e-cigarettes,” “it’s still tobacco,” “vaping”	Cost (*n* = 100)	“expensive,” “cheaper,” “costly,” “price,” “waste of money,” “cheaper in the long run”
5	Taste (*n* = 70)	“gross taste,” “bad taste in mouth,” “needs flavor,” “it’s tasty,” “minty”	Coolness (*n* = 48)	“Cool,” “socially unacceptable,” “silly,” “stupid,” “elegant”	Taste (*n* = 98)	“different flavors,” “great taste,” “bad taste,” “sweet,” “tastier”

* denotes categories that were not included in the close-ended belief options.

**Table 6 ijerph-18-00576-t006:** Multiple logistic regression adjusted odds ratio of any intention to use e-cigarettes, snus, or heated tobacco, *n* = 864.

	E-Cigarettes		Snus		Heated Tobacco	
	Self (*n* = 553)	Comparison (*n* = 275)	Self (*n* = 547)	Comparison (*n* = 268)	Self (*n* = 541)	Comparison (*n* = 274)
Beliefs	aOR [CI]		aOR [CI]		aOR [CI]	
Taste good	1.56 * [1.15 to 2.11]	2.33 * [1.43 to 3.78]	1.45 * [1.01 to 2.08]	1.51 [0.97 to 2.35]	2.92 * [1.91 to 4.47]	1.65 [0.87 to 3.12]
Feel harsh	0.97 [0.69 to 1.37]	0.66 [0.38 to 1.13]	0.64 * [0.43 to 0.96]	0.78 [0.43 to 1.42]	1.05 [0.71 to 1.55]	0.75 [0.37 to 1.53]
Odorless	0.93 [0.70 to 1.25]	0.85 [0.55 to 1.30]	1.40 [0.99 to 1.99]	1.41 [0.86 to 2.31]	0.78 [0.52 to 1.18]	0.91 [0.56 to 1.48]
Easy to use	1.42 [1.00 to 2.03]	1.54 [0.94 to 2.51]	1.22 [0.83 to 1.79]	1.15 [0.71 to 1.84]	1.25 [0.87 to 1.82]	1.26 [0.75 to 2.12]
Look cool /make me look cool	1.42 * [1.09 to 1.86]	1.01 [0.65 to 1.56]	1.24 [0.90 to 1.71]	1.57 [0.99 to 2.48]	1.25 [0.92 to 1.70]	1.92 * [1.19 to 3.09]
No shs made	1.26 [0.95 to 1.66]	1.32 [0.85 to 2.06]	0.68 * [0.47 to 0.97]	0.64 [0.39 to 1.03]	1.17 [0.81 to 1.67]	1.51 [0.94 to 2.43]
Risky	0.80 [0.56 to 1.14]	0.71 [0.42 to 1.19]	0.63 * [0.42 to 0.94]	1.30 [0.75 to 2.26]	0.88 [0.62 to 1.25]	0.58 [0.32 to 1.04]
Not addictive	0.91 [0.70 to 1.19]	1.32 [0.81 to 2.13]	2.07 * [1.50 to 2.85]	0.90 [0.57 to 1.42]	0.97 [0.68 to 1.40]	1.21 [0.79 to 1.84]
Long-term health benefits	1.14 [0.93 to 1.39]	0.97 [0.67 to 1.40]	1.03 [0.78 to 1.36]	1.13 [0.79 to 1.63]	1.39 * [1.09 to 1.76]	1.39 [0.92 to 2.10]
Cause lung damage/get lung damage	0.86 [0.58 to 1.27]	1.02 [0.62 to 1.68]	0.91 [0.64 to 1.29]	1.25 [0.77 to 2.04]	0.78 [0.55 to 1.10]	0.82 [0.50 to 1.35]
Has nicotine	0.79 [0.57 to 1.09]	0.76 [0.44 to 1.32]	1.02 [0.69 to 1.51]	0.50* [0.28 to 0.90]	0.91 [0.65 to 1.29]	1.42 [0.76 to 2.64]
Quit aid	1.61 * [1.23 to 2.11]	1.52 [0.97 to 2.38]	1.57 * [1.07 to 2.31]	1.93 * [1.19 to 3.11]	1.72 * [1.21 to 2.44]	0.83 [0.52 to 1.31]
Untrustworthy product science	0.61 * [0.46 to 0.81]	0.60 * [0.38 to 0.94]	0.59 * [0.41 to 0.84]	0.69 [0.44 to 1.08]	0.76 [0.55 to 1.05]	0.53 * [0.32 to 0.87]
Kid appeal	1.29 [0.97 to 1.72]	1.00 [0.67 to 1.51]	0.87 [0.63 to 1.21]	0.87 [0.56 to 1.35]	1.37 [1.02 to 1.85]	0.98 [0.63 to 1.52]
Expensive	0.99 [0.75 to 1.31]	0.75 [0.48 to 1.17]	1.20 [0.82 to 1.74]	1.33 [0.73 to 2.45]	0.67 * [0.47 to 0.96]	1.26 [0.77 to 2.06]
Demographic controls						
Ever MRTP use	4.93 * [2.78 to 8.76]	2.82 [1.22 to 6.54]	3.08 * [1.56 to 6.07]	6.27 * [2.66 to 14.79]	4.31 * [2.38 to 7.80]	2.62 * [1.10 to 6.24]
Current smoker	5.12 * [2.86 to 9.15]	17.94 * [6.71 to 47.92]	4.77 * [2.49 to 9.16]	5.70 * [2.29 to 14.18]	10.57 * [5.64 to 19.79]	3.95 * [1.82 to 8.60]
White	3.37 *[ 1.56 to 7.28]	0.57 [0.14 to 2.39]	1.38 [0.62 to 3.04]	1.19 [0.41 to 3.40]	0.89 [0.41 to 1.93]	2.49 [0.89 to 6.94]
Age in years	0.96 * [0.95 to 0.98]	0.97 [0.94 to 1.00]	0.97 * [0.95 to 0.99]	0.97 [0.95 to 1.00]	0.98 * [0.96 to 0.99]	0.97 * [0.95 to 1.00]
Hispanic	1.02 [0.36 to 2.87]	2.42 [0.19 to 30.72]	3.29 * [1.03 to 10.50]	0.64 [0.16 to 2.55]	1.01 [0.33 to 3.07]	2.92 [0.74 to 11.59]
Any college	1.47 [0.83 to 2.63]	2.16 [0.88 to 5.32]	1.35 [0.72 to 2.53]	1.66 [0.76 to 3.64]	1.77 [0.99 to 3.18]	1.24 [0.56 to 2.77]
Pseudo R^2^ (all variables)	0.50	0.57	0.51	0.42	0.50	0.41
Pseudo R^2^ (controls only)	0.33	0.40	0.25	0.26	0.30	0.24
Likelihood-ratio test	126.0 *	65.7 *	169.8 *	51.6 *	142.8 *	61.0 *

**Note.** Outcome was any intention to use the tobacco product. Adjusted models control for all variables in the table. Demographic control variables are dichotomous except for age. Likelihood-ratio test reports the LR chi2 difference between the full model and the model with controls only. Missing data were handled with listwise deletion and ranged from 2% to 6%. aOR = adjusted odds ratio. CI = 95% confidence interval. * *p* < 0.05.

## Data Availability

Deidentified data will be made available upon written request to the corresponding author after a data use agreement has been signed.

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
