# Peer review of "Harm Perceptions and Beliefs about Potential Modified Risk Tobacco Products"

_ijerph, 2021, doi:10.3390/ijerph18020576_

Round 1
Reviewer 1 Report
Comments for the author(s)
Thank you for having the opportunity to review the work entitled “Harm perceptions and beliefs about potential modified risk tobacco products”.
The subject is of great interest because of the importance of knowing aspects associated with the consumption of tobacco and other derived products. The impact on health systems and global health is unquestionable. Knowing behavioural, perceptual and of belief aspects that encourage individuals to consume it is of great interest.
Although the work is interesting and presents contributions to its field of study, it presents some structural and of content problems that can be corrected.
Next I will comment suggestions that can improve the work.
Introduction:
- The introduction must be improved. Although aspects and antecedents of legislation and the situation in his/her country are reviewed, there is not much bibliographic review on specific aspects that are studied in the work. The authors are studying awareness, harm perceptions, tobacco use and MRTP beliefs, etc. For this reason, more and in-depth information must be provided on the scientific evidence and what we know so far about these variables.
- At the end of the introduction, the authors must detail the objectives of the study and even hypotheses about the expected findings. This must be built on the basis of studies and previous evidence on the topic and the specific variables being studied. The objectives are a very important section because the work must revolve around them: the introduction must justify its proposal and approach, the methodology must allow them to be achieved, the results will show whether the objectives have been achieved or not and the discussion must argue to what extent they have been achieved, why or why they have not been achieved, if the results are in agreement or not with the previous evidence and if there have been problems or limitations when reaching the objectives. Later I will insist on this aspect.
Methods
- In the procedure the establishment of the two conditions is not clear, (one of them called "control") since it is later stated that "There were no differences between the conditions, so the conditions were collapsed for further analysis". Perhaps it is a problem of writing the text. The authors should clarify this point a bit more. There are or are not two control conditions and based on what they are established and for what purpose.
- In the same section it is stated that the “participants answered questions about awareness, harm perceptions, use, likelihood of use, beliefs, and intentions to use the three products that have potential for, or have already received, authorization for modified risk tobacco product claims ". The authors should expose in this section more details of the procedure for collecting these data, so that this study could be replicated in the future. Furthermore, the objectives pursued with these evaluations and what data were specifically collected are unknown. This makes it necessary to state the study objectives in advance.
- Regarding the measures / instruments section, the objectives of the work must be consistent with the instruments that are going to be used. The justification for these evaluation measures is not stated. But this section also presents some problems that the authors have to address:
- Awareness. The procedure followed might not be valid and/or reliable to establish that a subject is aware or not. “Participants were provided with brief descriptions e-cigarettes, snus and heated tobacco and then indicated if they had ever heard of each product. Those who answered yes were considered aware of the product. " You have to make a stronger justification that this measure is valid for asses what it is intended to measure.
- In relation to the MRTP beliefs survey, the authors must offer more data on its reliability and validity, provide references to other articles where its psychometric qualities are found, or highlight the lack of them as a limitation in the discussion section. The data provided on this validity and reliability are not sufficient. If the validity of the survey is questioned, the results can be questioned.
- If the authors are going to study aspects such as awareness, harm perceptions, tobacco use and MRTP beliefs, the introduction should provide more information and in greater depth about the previous scientific evidence and what we know so far about these aspects in relation to the consumption of tobacco or other similar products.
Statistic analysis:
- In the statistical analysis section, research questions should not be included. This section should only include a brief summary of the statistical analyses carried out. There are analyses included in the work (such as Spearman ranked correlations, descriptive analyses including means, SD, %, etc.) that must be detailed here.
Results:
- The results section is very ambitious by proposing too many sections. Being structured in sections, these could be objectives of the article and expressed at the end of the introduction section, and in any case, they must be justified as indicated earlier in the introduction.
Discussion:
- Authors should expand this section by commenting on the different groups of results. In relation to them, explain those results in detail, comment if they are in line with previous studies on the topic (or similar subjects of consumption of products harmful to health), if they agree with what is known about the topic until now and in case they differ from the previous evidence, present explanations of why this happens.
- In this section, limitations and problems of the study should be discussed. As limitations of the study, apart from those already indicated on the reliability and validity of the instruments, the authors should point out that the sample of participants is too small considering that the population it intends to represent is the entire United States population. This compromises the generalizability of the results.
In the conclusion section:
- Authors should be cautious and prudent in drawing conclusions based on their results due to the limitations of their work. The need for more future studies with larger samples should be noted.
Thanks for everything. I encourage the authors to continue working in this line.
Comentarios para los autores
Gracias por tener la oportunidad de revisar el trabajo titulado “Harm perceptions and beliefs about potential modified risk tobacco products”.
El tema es de gran interés por la importancia que tiene conocer aspectos asociados al consumo de tabaco y otros productos derivados. El impacto en los sistemas de salud y la salud global es incuestionable y conocer aspectos comportamentales, perceptivos y de creencias que incitan a los individuos a su consumo es de alto interés.
Aunque el trabajo es interesante y presenta aportes a su campo de estudio, presenta algunos problemas de estructura y contenido que pueden ser corregidos.
A continuación pasaré a comentar sugerencias de mejora sobre el trabajo.
Introducción:
- La introducción debe ser mejorada. Aunque se revisan aspectos y antecedentes de legislación y la situación en su país, no hay demasiada revisión bibliográfica sobre aspectos específicos que se estudian en el trabajo. Los autores van a estudiar awareness, harm perceptions, tabaco use and MRTP beliefs, etc. Por ello ha de aportarse más información y con más profundidad sobre la evidencia científica y lo que conocemos hasta ahora sobre estas variables.
- Al final de la introducción los autores deben detallar los objetivos del estudio e incluso hipótesis sobre los hallazgos esperados en base a estudios y evidencia previa sobre el tema y las variables específicas que se estudian. Los objetivos son un apartado muy importante porque sobre ellos debe girar el trabajo: la introducción debe justificar su propuesta y planteamiento, la metodología debe permitir alcanzarlos, los resultados demostrarán si se han alcanzado o no y la discusión debe argumentar en qué medida se han alcanzado, por qué o por qué no se han alcanzado, si los resultados están de acuerdo o no con la evidencia previa y si ha habido problemas o limitaciones a la hora de alcanzar los objetivos. Más adelante insistiré sobre este aspecto.
Procedimiento
- En el procedimiento no queda claro el establecimiento de las dos condiciones, una de ellas llamada “control” si luego se afirma que “There were no differences between the conditions, so the conditions were collapsed for further analysis”. Los autores deben aclarar un poco más este punto. Hay o no dos condiciones de control y en base a qué se establecen y con qué objetivo.
- En el mismo apartado se afirma que los aprticipantes “participants answered questions about awareness, harm perceptions, use, likelihood of use, beliefs, and intentions to use the three products that have potential for, or have already received, authorization for modified risk tobacco product claims”. Los autores deben exponer en este apartado más detalles del procedimiento de recolección de estos datos, de manera que este estudio pudiese ser replicado en el futuro. Además, se desconocen los objetivos que se persiguen con estas evaluaciones y datos específicamente recogidos. Esto hace necesario exponer con antelación los objetivos del estudio.
- Por lo que concierne al apartado de medidas/instrumentos, los objetivos del trabajo han de ser consistentes con los instrumentos que se van a utilizar. No queda expresada la justificación de estas medidas de evaluación. Pero además este apartado presenta algunos problemas que los autores han de abordar:
- El procedimiento seguido podría no ser válido ni fiable para establecer que un sujeto es o no aware. “Participants were provided with brief descriptions e-cigarettes, snus and heated tobacco and then indicated if they had ever heard of each product. Those who answered yes were considered aware of the product.” Hay que hacer una justificación más sólida de que esa medida es válida para asses lo que se pretende medir.
- En relación con el MRTP beliefs survey, los autores deben ofrecer más datos sobre su fiabilidad y validez, aportar referencias de otros artículos donde se encuentren sus cualidades psicométricas o destacar la inexistencia de las mismas como limitación en el apartado de discusión. Los datos aportados sobre este la validez y fiabilidad no son suficientes. Si se pone en cuestión la validez del survey, los resultados pueden ser cuestionados.
- Si los autores van a estudiar aspectos tales como awareness, harm perceptions, tabaco use and MRTP beliefs, en la introducción debe aportarse más información y con más profundidad sobre la evidencia científica previa y lo que conocemos hasta ahora sobre estos aspectos.
Análisis estadístico:
- En el apartado de análisis estadístico no debe incluirse research questions. Este apartado sólo debe incluir un breve resumen de los análisis llevados a cabo. Hay análisis incluidos en el trabajo (como por ejemplo Spearman ranked correlations, análisis descriptivos incluyendo medias, SD, %, etc.) que se han de detallar aquí.
Resultados:
- El apartado de resultados es muy ambicioso al proponer demasiados apartados. Al estar estructurado en apartados, estos podrían ser objetivos del artículo y expresarse al final del apartado de introducción, y en todo caso, han de ser justificados tal como se ha indicado más atrás, en la introducción.
Discusión:
Los autores deben ampliar este apartado haciendo comentarios a los distintos grupos de resultados. En relación con ellos, explicar esos resultados en detalle, comentar si van en la línea de estudios previos sobre el tema (o temas similares de consumo de productos dañinos para la salud), si están de acuerdo con lo que se sabe sobre el tema hasta ahora y en caso de que difieran de la evidencia previa, exponer explicaciones de por qué ocurre eso.
En este apartado deben comentarse limitaciones y problemas del estudio. Como limitaciones del estudio, aparte de las ya señaladas sobre la fiabilidad y validez de los instrumentos, los autores deben señalar que la muestra de participantes es demasiado pequeña teniendo en cuenta que la población a la que pretende representar es a toda la población de Estados Unidos. Esto compromete la generabilidad de los resultados.
En el apartado de conclusión:
Los autores deben ser cautos y prudentes con el establecimiento de conclusiones en base a sus resultados debido a las limitaciones de su trabajo. Debe señalarse la necesidad de más estudios futuros con muestras más grandes.
Gracias por todo y ánimo y suerte a los autores.
Author Response
International Journal of Environmental Research and Public Health
Manuscript Title: Harm perceptions and beliefs about potential modified risk tobacco products
Our letter responds to the reviewer’s helpful suggestions. The revisions to the paper have certainly improved it. Reviewer comments appear in italics below followed by our response. Changes in the revised paper are highlighted in yellow. We look forward to your publication decision.
Reviewer 1
Thank you for having the opportunity to review the work entitled “Harm perceptions and beliefs about potential modified risk tobacco products”. The subject is of great interest because of the importance of knowing aspects associated with the consumption of tobacco and other derived products. The impact on health systems and global health is unquestionable. Knowing behavioural, perceptual and of belief aspects that encourage individuals to consume it is of great interest. Although the work is interesting and presents contributions to its field of study, it presents some structural and of content problems that can be corrected.
Thank you for these kind comments.
1.1 The introduction must be improved. Although aspects and antecedents of legislation and the situation in his/her country are reviewed, there is not much bibliographic review on specific aspects that are studied in the work. The authors are studying awareness, harm perceptions, tobacco use and MRTP beliefs, etc. For this reason, more and in-depth information must be provided on the scientific evidence and what we know so far about these variables.
We have updated the introduction to include a more compelling rationale to study beliefs as the antecedents of behavior. “The potential impact of MRTP claims on population health may in part depend on whether smokers are exposed to these claims, how they view MRTP claims on their own and in comparison to other tobacco products, and if they perceive them to be salient and truthful. [24–26] Using a theory of reasoned action lens to approach this topic, how the population reacts to these claims will be in part determined by their beliefs (including misbeliefs) about using these products. [27] Health mass media campaigns that follow principles of effective campaigns can have moderate effects on health knowledge, beliefs, attitudes, and behaviors. [28,29] One of these principles is helping campaign designers understand the nature of the behavior, for a campaign can only be effective if the beliefs targeted by the campaign impact the intended behavior.”
1.2 At the end of the introduction, the authors must detail the objectives of the study and even hypotheses about the expected findings. This must be built on the basis of studies and previous evidence on the topic and the specific variables being studied. The objectives are a very important section because the work must revolve around them: the introduction must justify its proposal and approach, the methodology must allow them to be achieved, the results will show whether the objectives have been achieved or not and the discussion must argue to what extent they have been achieved, why or why they have not been achieved, if the results are in agreement or not with the previous evidence and if there have been problems or limitations when reaching the objectives. Later I will insist on this aspect.
We have moved the objectives section into the introduction section and expanded them with more details that can help the reader understand our rationale and justify the approach and provide a framework to aid the reader in understanding the results we present later in the paper. “The purpose of this this study was to provide valuable baseline data about smokers’ and former smokers’ beliefs about three products that currently have been or have the potential to be authorized as modified risk tobacco products. Specifically, we sought to understand harm perceptions and beliefs about products with the potential to be authorized as MRPTs; examine differences in beliefs about using MRTPs and beliefs about MRTPs in comparison to cigarettes; how MRTP user status impacts the rating of beliefs; and which beliefs are associated with intentions to use MRTPs.”
1.3 In the procedure the establishment of the two conditions is not clear, (one of them called "control") since it is later stated that "There were no differences between the conditions, so the conditions were collapsed for further analysis". Perhaps it is a problem of writing the text. The authors should clarify this point a bit more. There are or are not two control conditions and based on what they are established and for what purpose.
Thank you for pointing our misstep in labeling two different conditions generic. We have removed the word generic from the description of the information condition and used it only for describing the wording condition. This section of the paper has been edited to state: “Participants were randomly assigned to either the control condition, in which they read a description about heat-not burn tobacco, snus, and e-cigarettes, presented in random order; or to the corporate social responsibility condition, in which they read the generic description, plus a corporate responsibility statement crafted using press releases and text from IQOS, General Snus, or JUUL’s website respectively. There were no differences between the conditions, so the conditions were collapsed for further analysis.”
1.4 In the same section it is stated that the “participants answered questions about awareness, harm perceptions, use, likelihood of use, beliefs, and intentions to use the three products that have potential for, or have already received, authorization for modified risk tobacco product claims ". The authors should expose in this section more details of the procedure for collecting these data, so that this study could be replicated in the future. Furthermore, the objectives pursued with these evaluations and what data were specifically collected are unknown. This makes it necessary to state the study objectives in advance.
Our objectives are now stated in detail at the end of the introduction section, which provides the rationale for each of the measures (further details in response to comment 1.2). The measures for each of the items in the procedures section are now further detailed in the measures section to ensure other can easily replicate our study. These edits can be found in table 2, appendix A, and section 2.2.
1.5 Regarding the measures / instruments section, the objectives of the work must be consistent with the instruments that are going to be used. The justification for these evaluation measures is not stated. But this section also presents some problems that the authors have to address: Awareness. The procedure followed might not be valid and/or reliable to establish that a subject is aware or not. “Participants were provided with brief descriptions e-cigarettes, snus and heated tobacco and then indicated if they had ever heard of each product. Those who answered yes were considered aware of the product. " You have to make a stronger justification that this measure is valid for asses what it is intended to measure.
Our measures come or were adapted from prior research in the area, including a large federally funded national study Population Assessment of Tobacco and Health (PATH). We have updated this section (section 2.2) with appropriate citations included after each measure.
1.6 In relation to the MRTP beliefs survey, the authors must offer more data on its reliability and validity, provide references to other articles where its psychometric qualities are found, or highlight the lack of them as a limitation in the discussion section. The data provided on this validity and reliability are not sufficient. If the validity of the survey is questioned, the results can be questioned.
We have provided additional rational about the set of beliefs we tested “The decision to focus on these beliefs was based on prior qualitative research, surveys on salient beliefs, and examining the MRTP applications and materials made publicly available. [12,25,34–42]”
1.7 If the authors are going to study aspects such as awareness, harm perceptions, tobacco use and MRTP beliefs, the introduction should provide more information and in greater depth about the previous scientific evidence and what we know so far about these aspects in relation to the consumption of tobacco or other similar products.
The constructs we have used in this paper were chosen using a Theory of Reasoned Action lens. Demographic and social factors shape beliefs, which influence attitudes, which in turn impact health beliefs. The introduction now includes the following paragraphs.
“In response to health concerns about the harmful health effects of smoking, the tobacco industry began marketing new tobacco products as less harmful compared to traditional cigarettes. [13,14] This history can be traced from the recent efforts to advertise e-cigarettes as a harm reduction alternative to cigarettes, before becoming regulated by the FDA, [15,16] back to falsely claiming that filtered, “low tar,” and “light” cigarettes were less harmful than regular cigarettes. [17–19]”
The potential impact of MRTP claims on population health may in part depend on whether smokers are exposed to these claims, how they view MRTP claims on their own and in comparison to other tobacco products, and if they perceive them to be salient and truthful. [24–26] Using a theory of reasoned action lens to approach this topic, how the population reacts to these claims will be in part determined by their beliefs (including misbeliefs) about using these products. [27] Health mass media campaigns that follow principles of effective campaigns can have moderate effects on health knowledge, beliefs, attitudes, and behaviors. [28,29] One of these principles is helping campaign designers understand the nature of the behavior, for a campaign can only be effective if the beliefs targeted by the campaign impact the intended behavior.
1.8 In the statistical analysis section, research questions should not be included. This section should only include a brief summary of the statistical analyses carried out. There are analyses included in the work (such as Spearman ranked correlations, descriptive analyses including means, SD, %, etc.) that must be detailed here.
We have removed the research questions from this section and provided more details about these statistical methods used (section 2.3).
1.9 The results section is very ambitious by proposing too many sections. Being structured in sections, these could be objectives of the article and expressed at the end of the introduction section, and in any case, they must be justified as indicated earlier in the introduction.
These edits have been incorporated and are further detailed in the responses to comments 1.1, 1.2, and 1.7.
1.10 Authors should expand this section by commenting on the different groups of results. In relation to them, explain those results in detail, comment if they are in line with previous studies on the topic (or similar subjects of consumption of products harmful to health), if they agree with what is known about the topic until now and in case they differ from the previous evidence, present explanations of why this happens.
- In this section, limitations and problems of the study should be discussed. As limitations of the study, apart from those already indicated on the reliability and validity of the instruments, the authors should point out that the sample of participants is too small considering that the population it intends to represent is the entire United States population. This compromises the generalizability of the results.
We have rewritten our limitations into a separate paragraph and included the convience sample as a limitation to generalizability. It now reads, “One of the limitations of this study is that it was a convenience sample that cannot be generalized to the larger US population. While the demographics are similar to those of US smokers, our population was slightly more educated than the population of US smokers. Another limitation is that it is a cross-sectional study and only captures these beliefs at a very specific point in time. As previously mentioned, these data were collected before EVALI news was widespread, and before a global pandemic featuring a respiratory illness (COVID-19) interrupted everyone’s daily lives and routines. While it is encouraging that some of our findings are similar to other large nationally representative longitudinal studies, [36,50,52] replicability of these findings with a larger and longitudinal sample would be ideal.”
In the conclusion section:
- Authors should be cautious and prudent in drawing conclusions based on their results due to the limitations of their work. The need for more future studies with larger samples should be noted.
We have moved the limitation paragraph, which mentions the need for a larger sample, so that it immediately precedes the conclusions section. The conclusion section has been edited to read “For all three products, believing that MRTPs taste good and can be used as a quit aid are associated with greater intentions to try MRTPs. This is consistent with prior work which found that young adults who held a baseline belief that e-cigarettes could help people quit were more likely to report experimenting with e-cigarettes at follow-up.[39] Beliefs that the science about the product was untrustworthy was associated with reduced intentions to try the product. However, each product had its own unique sets of beliefs associated with trying the products. This paper contributes to the growing literature about beliefs and perceptions of e-cigarettes, snus, and heated tobacco. It contributes to our understanding of what the public believes about products currently or potentially authorized to be marketed as modified risk tobacco products. This understanding can inform the planning and development stages of communication campaigns.”
Reviewer 2 Report
See attached Pdf.
Line 16: I suggested expanding the meaning of SNUS as this is not a common vocabulary and to make the paper more pleasurable to read [Snus is a moist powder smokeless tobacco product
I think readers will enjoy having all information in one place as much as possible. snus is not a familiar word so a short sentence indicating what it is makes the paper easier to read. My opinion.] Line 42: I think the introduction will benefit from inclusion of the meaning of IQOS which is shortcut for "I Quit Ordinary Smoking" As it is readers will wonder what that abbreviation stands for. Line 46: The "In Oct, 2019." is not a complete sentence. please consider reviewing how the sentences stand together. And why use "Oct" when you later use "July" consistency helps.I suggest writing October, 2019; .... Line 50: Need to join two the ... claims, and perceive ..." Another option depending on your view is "... claims, or perceive ..." Line 51: their misbeliefs is also their belief. I think the statement "...determined by their beliefs about these products" should be enough Line 57: what is EVALI, many readers are not familiar with this term. It will help if what EVALI stands for is included (note that you expand the meaning of SSI but ot EVALI, why?) Line 65: Lost here "ever"? you mean "never users of ..." Line 227: think the sentences will be easier to read if we use the term "rated" instead of "endorse" It is easier to understand that something is rated second and third; than saying endorsed 2nd and third. Also, sentences like "... was most often rated second and third" appear less confusing than ".... rated second and third most often." Line 213: "... heated tobacco would ..." would be more appropriate?
Line 227: "... belief that, compared to cigarettes, heated tobacco ..." Please, Note the punctuation Line 288-289: "... endorsements of the 15 beliefs about e-cigarettes and heated tobacco" This read better?

Author Response
International Journal of Environmental Research and Public Health
Manuscript Title: Harm perceptions and beliefs about potential modified risk tobacco products
Our letter responds to the reviewer’s helpful suggestions. The revisions to the paper have certainly improved it. Reviewer comments appear in italics below followed by our response. Changes in the revised paper are highlighted in yellow. We look forward to your publication decision.
2.1 Line 16: I suggested expanding the meaning of SNUS as this is not a common vocabulary and to make the paper more pleasurable to read Snus is a moist powder smokeless tobacco product
I think readers will enjoy having all information in one place as much as possible. snus is not a familiar word so a short sentence indicating what it is makes the paper easier to read. My opinion.
We have included a description of each product in table 2.
2.2 Line 42: I think the introduction will benefit from inclusion of the meaning of IQOS which is shortcut for "I Quit Ordinary Smoking" As it is readers will wonder what that abbreviation stands for.
While it is true that this is what IQOS stands for, we are using it here as an a brand name, an example of heated tobacco, and the majority of the advertisements do not include this as an acronym and we feel it’s common usage is in the word itself and not what it stands for. Similarly, we would not define GEICO as the Government Employee Insurance Company.
Line 46: The "In Oct, 2019." is not a complete sentence. please consider reviewing how the sentences stand together. And why use "Oct" when you later use "July" consistency helps.I suggest writing October, 2019; ....
We have corrected this typo and stylistic error.
Line 50: Need to join two the ... claims, and perceive ..." Another option depending on your view is "... claims, or perceive ..." Line 51: their misbeliefs is also their belief. I think the statement "...determined by their beliefs about these products" should be enough
We have edited this section so that it now reads “The potential impact of MRTP claims on population health may in part depend on whether smokers are exposed to these claims, how they view MRTP claims on their own and in comparison to other tobacco products, and if they perceive them to be salient and truthful. [24–26] Using a theory of reasoned action lens to approach this topic, how the population reacts to these claims will be in part determined by their beliefs (including misbeliefs) about using these products. [27]”
Line 57: what is EVALI, many readers are not familiar with this term. It will help if what EVALI stands for is included (note that you expand the meaning of SSI but ot EVALI, why?)
We have updated this to include the definition of EVALI.
Line 65: Lost here "ever"? you mean "never users of ..."
We were including ever-users of e-cigarettes to mean current or former users of e-cigarettes. We have edited it for clarity, it now reads “A little more than half of the participants were current smokers or ever-users (i.e. current or former users) of e-cigarettes”.
Line 227: think the sentences will be easier to read if we use the term "rated" instead of "endorse" It is easier to understand that something is rated second and third; than saying endorsed 2nd and third. Also, sentences like "... was most often rated second and third" appear less confusing than ".... rated second and third most often."
We agree that this simplifies the reading and have made these changes throughout the paper (sections 3.3.3 and 3.5.3).
Line 213: "... heated tobacco would ..." would be more appropriate?
We have made this edit. (section 3.5.1)
Line 227: "... belief that, compared to cigarettes, heated tobacco ..." Please, Note the punctuation
This now reads “For the belief that heated tobacco would not create second-hand smoke compared to cigarettes, current users and former users rated it 2nd and 3rd highest, respectively (Mcurrent=3.33 , SD=1.01; Mformer=3.21 , SD=1.00). In comparison, never-users rated the belief 9th highest (M=2.84 , SD=.89).”
Line 288-289: "... endorsements of the 15 beliefs about e-cigarettes and heated tobacco" This read better?
This improves the readability and we have made the suggested edit.
Reviewer 3 Report
The authors have conducted an interesting survey on smokers and users of modified risk products. The primary objective of the paper is to ascertain if current or past smokers are influenced in their intention to use alternative nicotine delivery systems in the future as a result of holding specific beliefs concerning the characteristics of the product. They run logistic regressions to determine the odds ratios associated with different characteristics/variables.
They find that the two strongest determinants of use intentions are whether the individuals are current smokers (or not) and whether they have ever experimented with modified risk tobacco products (or not). They also include in their logistic regressions 15 variables that define the characteristics of the three different modified risk products being examined. They find that a subset of these 15 variables are significant in the statistical sense.
I have some concerns on the interpretation of the coefficients, and the paper would benefit from more work in certifying that the claims in the paper are statistically correct. Consider the following thought experiment.
Imagine that decisions are being made in three consecutive time periods - the past, the present and the future. All individuals in the sample smoked in the past, and about half of them used modified risk products in addition. What determined the decision to smoke or use a modified risk product in the past? It is difficult to think of any influence that is not included in the 15 variables used by the authors. That is to say, the status variables “current smoker” and “ever used modified risk products” are themselves determined by those 15 variables. As a consequence, we are effectively using those 15 variables twice, once, in the past, in determining whether the individual had smoked or not (or used a modified risk product) and again, today, in a second regression. Hence it is not clear what the coefficients on each of the 15 regressors really mean.
As a first check the authors could perform a test on the joint significance of all 15 variables: Can we sure statistically that they add explanatory power to the regression beyond the two critical (smoking and modified product use) variables plus the demographic variables? In general, if we had a well-specified regression model and then added 15 variables, randomly chosen, we would likely find some of them significant, but (hopefully) would not find that they increase the explanatory power of the whole model.
Then the authors might help the reader to understand the results better by explaining the omitted categories. For example, if the person is not Hispanic, what is he or she? And what do we expect the constant to be – close to unity or zero? If we set all of the variables equal to their ‘alternative’ value (zero?), does that help us interpret the constant – I am uncertain myself here?
Virtually all the 15 survey variables are either insignificant or have coefficients just slightly different from unity where they are significant. This is another reason I worry that the group as a whole may have no significance. But if we find that the group does have significance, it would be of interest to experiment with dropping specific coefficients to observe how the remaining coefficients behave.
Author Response
International Journal of Environmental Research and Public Health
Manuscript Title: Harm perceptions and beliefs about potential modified risk tobacco products
Our letter responds to the reviewer’s helpful suggestions. The revisions to the paper have certainly improved it. Reviewer comments appear in italics below followed by our response. Changes in the revised paper are highlighted in yellow. We look forward to your publication decision.
Reviewer 3
3.1 The authors have conducted an interesting survey on smokers and users of modified risk products. The primary objective of the paper is to ascertain if current or past smokers are influenced in their intention to use alternative nicotine delivery systems in the future as a result of holding specific beliefs concerning the characteristics of the product. They run logistic regressions to determine the odds ratios associated with different characteristics/variables.
They find that the two strongest determinants of use intentions are whether the individuals are current smokers (or not) and whether they have ever experimented with modified risk tobacco products (or not). They also include in their logistic regressions 15 variables that define the characteristics of the three different modified risk products being examined. They find that a subset of these 15 variables are significant in the statistical sense.
I have some concerns on the interpretation of the coefficients, and the paper would benefit from more work in certifying that the claims in the paper are statistically correct. Consider the following thought experiment.
Imagine that decisions are being made in three consecutive time periods - the past, the present and the future. All individuals in the sample smoked in the past, and about half of them used modified risk products in addition. What determined the decision to smoke or use a modified risk product in the past? It is difficult to think of any influence that is not included in the 15 variables used by the authors. That is to say, the status variables “current smoker” and “ever used modified risk products” are themselves determined by those 15 variables. As a consequence, we are effectively using those 15 variables twice, once, in the past, in determining whether the individual had smoked or not (or used a modified risk product) and again, today, in a second regression. Hence it is not clear what the coefficients on each of the 15 regressors really mean.
As a first check the authors could perform a test on the joint significance of all 15 variables: Can we sure statistically that they add explanatory power to the regression beyond the two critical (smoking and modified product use) variables plus the demographic variables? In general, if we had a well-specified regression model and then added 15 variables, randomly chosen, we would likely find some of them significant, but (hopefully) would not find that they increase the explanatory power of the whole model.
Our paper is using the Theory of Reasoned Action as a lens for exploring how MRTP beliefs predict the intentions to use MRTPs. For decades, misleading claims from the tobacco companies have led people to wrongly believe that certain tobacco products (e.g. “light”, filtered, or low-tar cigarettes) were less harmful than “regular” cigarettes. When products come onto market with these MRTP authorizations, it will be important, for public health and for legal compliance, that these MRTP claims not mislead people in a similar fashion. Beliefs can be targeted as potential themes for communication campaigns, but the campaign will not have the desired effect if the beliefs targeted by the campaign are not linked to intentions and behaviors. Because these beliefs and intentions could be confounded by common causes (demographics, and prior MRTP use experience), we must take these into consideration before reporting the association between beliefs and intentions.
From your comments, I surmise that the model you are proposing is that prior demographics/experience à prior behavior à beliefs à intentions. It is true, that in trying this model we would want to use the residual belief scores on intentions. While this model is not uninteresting, we are most concerned with the last step in the model, and not the prior causal process. Because of this, we have chosen to examine how beliefs and intentions are associated after controlling for the exogenous predictors.
We have amended the introduction to expand up on our objectives and purpose, to make it clearer to the reader why we have chosen to focus on beliefs. It has been edited to include the following paragraphs “In response to health concerns about the harmful health effects of smoking, the tobacco industry began marketing new tobacco products as less harmful compared to traditional cigarettes. [13,14] This history can be traced from the recent efforts to advertise e-cigarettes as a harm reduction alternative to cigarettes, before becoming regulated by the FDA, [15,16] back to falsely claiming that filtered, “low tar,” and “light” cigarettes were less harmful than regular cigarettes. [17–19]”
“The potential impact of MRTP claims on population health may in part depend on whether smokers are exposed to these claims, how they view MRTP claims on their own and in comparison to other tobacco products, and if they perceive them to be salient and truthful. [24–26] Using a theory of reasoned action lens to approach this topic, how the population reacts to these claims will be in part determined by their beliefs (including misbeliefs) about using these products. [27] Health mass media campaigns that follow principles of effective campaigns can have moderate effects on health knowledge, beliefs, attitudes, and behaviors. [28,29] One of these principles is helping campaign designers understand the nature of the behavior. “
Additionally, we have labeled the table as beliefs, and demographic controls to help the reader clarify how we are thinking about the demographic variables in the regression equation.
3.2 Then the authors might help the reader to understand the results better by explaining the omitted categories. For example, if the person is not Hispanic, what is he or she? And what do we expect the constant to be – close to unity or zero? If we set all of the variables equal to their ‘alternative’ value (zero?), does that help us interpret the constant – I am uncertain myself here?
We have included a note in the table that “Demographic control variables are dichotomous except for age”. We have removed the constant from the table, as without a 0 point for the beliefs or age, it is largely meaningless to interpret.
Virtually all the 15 survey variables are either insignificant or have coefficients just slightly different from unity where they are significant. This is another reason I worry that the group as a whole may have no significance. But if we find that the group does have significance, it would be of interest to experiment with dropping specific coefficients to observe how the remaining coefficients behave.
It is true that many of the beliefs do not individually predict intentions to try a product. This is potentially good news for campaign designers as having a narrow set of beliefs that are associated with the behavior makes the design and pretesting process simpler and less costly.
Round 2
Reviewer 3 Report
This paper contains a very rich analysis of an interesting data basis. The study is informative and valuable. It is close to acceptance.
I am requesting one small addition to the analysis that can be performed in a couple of hours.
In the multivariate analysis I note that the results are now (nicely) divided into demographic variables and the information/belief variables. In order to both reinforce the validity and interpretation of the results in the text, the regressions should be run without any of the 15 belief variables, just including the demographic variables, and the pseudo R^2 should be calculated. This bare bones regression can then be compared to the richer regression with all 21 variables included. A test of statistical significance of the additional 15 variables can then be performed. Assuming these 15 are significant as a group, this would add validity to the interpretations.
Reporting the outcome of this experiment need only occupy a sentence or two.
Author Response
Thank you for these comments. We have performed the additional analyses. We have updated table 6. It now includes the Pseudo R2 for the controls only model and the results of the Likelihood-ratio test comparing the two models.
We have updated the Results section (line 272 and 273) and the Methods section (line 162-164) to reflect these analyses.
Thank you.
